# Using the Data-Compression Method for Studying Hunting Behavior in Small Mammals

**DOI:** 10.3390/e21040368

**Published:** 2019-04-04

**Authors:** Zhanna Reznikova, Jan Levenets, Sofia Panteleeva, Anna Novikovskaya, Boris Ryabko, Natalia Feoktistova, Anna Gureeva, Alexey Surov

**Affiliations:** 1Institute of Animal Systematics and Ecology, Siberian Branch of the Russian Academy of Sciences, Novosibirsk 630091, Russia; 2Novosibirsk State University, Novosibirsk 630090, Russia; 3Institute of Computational Technologies, Siberian Branch of the Russian Academy of Sciences, Novosibirsk 630090, Russia; 4Severtsov Institute of Ecology and Evolution, Russian Academy of Sciences, Moscow 119071, Russia

**Keywords:** data compression, complexity, biological texts, hunting stereotype, rodents, insectivorous

## Abstract

Using the data-compression method we revealed a similarity between hunting behaviors of the common shrew, which is insectivorous, and several rodent species with different types of diet. Seven rodent species studied displayed succinct, highly predictable hunting stereotypes, in which it was easy for the data compressor to find regularities. The generalist Norway rat, with its changeable manipulation of prey and less predictable transitions between stereotype elements, significantly differs from other species. The levels of complexities of hunting stereotypes in young and adult rats are similar, and both groups had no prior experience with the prey, so one can assume that it is not learning, but rather the specificity of the organization of the stereotype that is responsible for the nature of the hunting behavior in rats. We speculate that rodents possess different types of hunting behaviors, one of which is based on a succinct insectivorous standard, and another type, perhaps characteristic of generalists, which is less ordered and is characterized by poorly predictable transitions between elements. We suggest that the data-compression method may well be more broadly applicable to behavioral analysis.

“Though this be madness, yet there is method in it”Shakespeare’s *Hamlet*, 1602

## 1. Introduction

One of the main problems in studying animal behavior at different levels of organization, from individuals to collective behavior of organisms, is searching for a reliable criterion for evaluating the variability and complexity of behavioral patterns. In the field of comparative ethology, the study of the structure of behavior is based on identification of specific behavioral patterns [1,2], which, in turn, are viewed as classes defined by regularities in one or more of the following five domains: Location, orientation, physical topography, intrinsic properties, and physical effects [3]. Attempts to examine the organizational complexity of behavioral patterns have been aimed at different aspects of animal behavior, such as signal activities and communication [4,5,6,7,8,9], division of labor within ant communities [10], modifications of behavior under stress conditions [11], modifications of escape trajectories and predator evasion abilities in rodents [12], food-caching sequences in canine species [13], and so on. The prevalent method of comparative ethological studies is based on the analysis of ethograms, that is, recordings of behavioral sequences of letters from an alphabet that consists, on average, of 10–15 symbols or letters, each corresponding to a certain behavioral element (an act) [14,15]. We analyze ethograms as “biological texts”, that is, as sequences of symbols from a finite alphabet [16].

To compare the behavior of different species one needs to identify specific patterns in sequences of behavioral acts. A critical difficulty here is finding an adequate model which would allow for an assessment of specific characteristics of a “text” while using a relatively small number of parameters because large amounts of data are typically not available in behavioral applications.

The data-compression method suggested by Ryabko et al. [16] is based on the concept of Kolmogorov complexity and allows us to search for regularities within sequences of symbols using data compressors (see details in References [16,17,18]). The main idea behind the data compression method is that it is able to capture all kind of regularities in a “text”, and do so in a way amenable to formal statistical analysis. By regularity, we mean any characteristic of a text that makes it more predictable, such as repeating subsequences, letters or sequences that are more likely to occur when preceded or followed by certain other letters or subsequences, and so on. In general, regularities may be complex and involve arbitrary computable functions, but when applying a data compressor we restrict ourselves to those regularities it can capture. When comparing ethograms of different species as biological “texts”, the method implies the presence of regularities within behavioral patterns and finds all differences in the complexity of the ethograms. If we see such differences, then additional biological data are needed to explain them.

When applied to the members of different branches of animal kingdom, like seagulls, and ants, the data-compression method appears to be a useful tool to distinguish between “basic” stereotypical behavioral patterns and flexible behavior [16], and to reveal similarities between hunting behavior in the insectivorous common shrew and the granivorous striped field mouse [18].

In this study, we compare, for the first time, the organizational complexity of species-specific hunting stereotypes in nine species of small mammals with various diets and different levels of phylogenetic relationship. To describe behavior, we use the same notions as in Reference [17]: We denote elementary movements and postures as minimal units of behavior (‘‘behavioral elements’’, for brevity), we call a ‘‘behavioral sequence’’ an arbitrary sequence of successive behavioral elements, and a ‘‘behavioral stereotype’’—a relatively stable chain of behavioral elements. Note that the behavioral stereotype may be part of a behavioral sequence and may coincide with it completely. If the variability of a stereotype is low, and it is possible with high probability to predict the appearance of each element in it, then the stereotype elements are stably related to each other. We call such a stereotype succinct, and from the ethological point of view, it may qualify as rigid and deterministic. The stereotypes of courtship and of territorial defense can serve as good examples here, as well as hunting stereotypes of highly specialized predators [16,17]. The higher the variability of the stereotype, the less likely we are to predict transitions from one element to another. We call such a stereotype flexible and suggest optional hunting behavior in rodents as a good model.

We revealed for the first time a hunting behavior in four omnivorous hamster species and two herbivorous vole species. Including two rodent species that we studied before [18], we were able to compare the hunting behavior of 8 rodent species belonging to the super-family Muroidea, in which nearly all elements of hunting behavior were common, while the ratios in the behavioral sequences significantly differed. We used a specialized insectivorous species (the common shrew) as a ‘standard’ insect hunter. The use of the data-compression method allowed us to evaluate the flexibility of hunting stereotypes or their succinctness and speculate about different types of hunting behaviors in rodents.

## 2. Materials and Methods

### 2.1. Animals and Housing

The experiments were conducted in the laboratory in 2012–2018 on nine species of small mammals. We used 81 non-pedigree adult Norway rats *Rattus norvegicus* (41 females and 40 males) and 52 young rats naive to potential prey at 30–40 days of age (23 females and 29 males), 26 striped field mice *Apodemus agrarius* (13 females and 13 males), 19 Campbell’s dwarf hamster *Phodopus campbelli* (8 females and 11 males), 30 Djungarian hamster *P. sungorus* (14 females and 16 males), 8 Eversmann’s hamster *Allocricetulus eversmanni* (7 females and 1 males), 13 Mongolian hamster *Al. curtatus* (8 females and 5 males), 46 narrow-headed voles *Lasiopodomys gregalis* (23 females and 23 males), 53 Tuva silver vole *Alticola tuvinicus* (25 females and 28 males), 11 common shrews *Sorex araneus* (7 females and 4 males). All animals were housed in plastic cages containing cotton nesting material, they were fed each day once, and they had *ad libitum* access to water; all animals were provided with all types of food before being taken into experimental arenas (see details in Reference [18]). Out of the 26 mice, 9 (6 females and 3 males) were born in the laboratory as the progeny of the wild-caught mice, while 10 males and 7 females, as well as all of *S. araneus, L. gregalis,* and 4 individuals of *Alt. tuvinicus* were captured in the natural environment with the use of Sherman’s traps. Members of the other species were descendants caught in the wild and have not previously encountered live insects. We used imago and last instar nymphs of the lobster cockroach *Nauphoeta cinerea* (27.93 ± 0.40 mm) as live mobile prey. Insects were kept and bred in our laboratory in separate containers. The percentages of the prey’s body length versus the predator’s body length was 12% in *R. norvegicus*, 15% in *Al. eversmanni* and *Al. curtatus*, 23% in young *R. norvegicus*, *A. agrarius, L. gregalis*, *Alt. tuvinicus*, and 30% in *P. campbelli*, *P. sungorus* and in *S. araneus*.

All experiments with rodents were performed in accordance with the rules adopted by the European Convention for the Protection of Vertebrate Animals used for Experimental and other Scientific Purposes. The experimental protocol was approved by the Bioethical Committee of Severtsov Institute of Ecology and Evolution Russian Academy of Sciences (protocol 22, 3 May 2018).

### 2.2. Experimental Procedures

We placed each vertebrate animal in a separate Noldus arena: 45 × 45 × 50 cm for the adult rats and 30 × 30 × 35 for young rats and the other eight species. The arena was covered with a transparent lid in order to prevent animals from getting out. In each trial, an insect was placed into the arena manually, 5 min after the vertebrate animal. Video recordings were made using a Sony Handycam DCR-SR68 camera (frame rate, 25 frames per second, Sony Corporation, Sony City, Minato, Tokyo, Japan) for most rodent species, and a Sony HDR-AS200V (60 frames per second, Sony Corporation, Sony City, Minato, Tokyo, Japan) for *S. araneus, Al. eversmanni* and *Al. curtatus.* Animals received three insects in turn. In cases of unsuccessful hunting, we waited for 10 min since the last contact between the animal and the prey, and then finished the observation. After each test, the arena was cleaned using 70% alcohol. We selected complete hunting stereotypes for the analysis, that is, those which ended with killing and eating the prey.

### 2.3. Data Encoding

The alphabet for the analysis of hunting behaviors was devised based on the video recordings. Videos were slowed down 25 times from their normal speed, that is, with a temporal resolution of 1 frame per second for rodents and 2.4 for shrews. With the use of The Observer XT 10.1 (Noldus Information Technology b.v., Wageningen, The Netherlands), we selected 19 behavioral elements (see details in Reference [18]) and divided them into three groups. The first one includes “key” elements that are strongly necessary for accomplishing the hunting stereotype, such as bite (W), seizing an insect with paws (E) (observed in rodents only: Figure 1), pursuing the prey by walking (S) or running (Q). The second group includes “auxiliary” elements related to the prey handling (R), sniffing (D), carrying the prey in the teeth (G), nibbling the insect’s legs (H), and pinning it down to the ground with one (N) or two (M) paws (the latter two elements were observed in *S. araneus* only). The third group consisted of the “noise” elements which do not influence the performance of the stereotype at all: (C) freezing, (V) turning a body to 90°, (B) U-turn, (F) turning of the head, (Y) rearing against the wall, (I) free-standing rearing, (U) backwards movement, (X) self-grooming, and (J) jump. We included “J” into the third group of elements because our animals do not jump at an insect; they just jump up on the spot.

Video recordings were processed by assigning a letter to each of the elements of behavior, in the order of their appearance, without taking into account their duration. This results in a sequence of letters. For example, if an animal stalks a prey for 10 s and then bites, the sequence would be QW. If an animal committed a behavioral act several times in a row, we recorded this as follows: One bite—W, three bites—WWW, a bite and two captures with paws—WEE.

### 2.4. Hypothesis Testing 

The essence of the data compression method [16] is that we are trying to apply an adequate model which would allow for an assessment of certain characteristics of a text while using a relatively small number of parameters. This section of the methods follows very closely the explanation provided in the corresponding sections of [18]. The degree of complexity of a “text” could be estimated by its Kolmogorov complexity. Although Kolmogorov complexity is not algorithmically computable, it can be, in a certain sense, estimated by means of data compressors. The data compression method allows us to combine the advantages of methods based on Kolmogorov complexity with classic methods of testing statistical hypotheses.

We consider the two following hypotheses: *H*_0_ = {the sequences from both sets are generated by one source} and *H*_1_ = {the sequences from the different sets are generated by stationary and ergodic sources with different Kolmogorov complexities per letter of generated sequences}. Specifically, this can be done as follows: (1) From the sequences to be compared fragments (*x*_1_…*x_t_*) of equal length *t* are selected randomly so that the Mann–Whitney–Wilcoxon test can be applied to the resulting fragments; (2) the complexity of each fragment is defined as *K*(*x*_1_…*x_t_*) = |*ϕ*(*x*_1_…*x_t_*)|/*t*, where *ϕ* is a data compressor, and |*ϕ*(*x*_1_…*x_t_*)| is the length of the fragment of the sequence compressed by the data compressor; (3) applying the Mann–Whitney–Wilcoxon test we test the Hypothesis *H*_0_. To test the hypotheses, we should represent the sequence of symbols as text files. Then these text files should be compressed by the chosen data compression method. The level of compression corresponds to the ratio between the length of the file after and before the compression. The difference between compression ratios of files to be compared reflects the difference between the complexities of the symbol sequences recorded. So we can use the compression ratio as a characteristic of complexity (see details in Reference [18]).

The possibility to compress sequences (or “texts”) realized by different data compressors is highly dependent on the chosen method of compression, that is, on the algorithm used to find regularities in the file to be compressed. The more regularities the compression method used can spot, the better is the compression and thus the smaller is the estimated complexity of the text in the compressed file, and, finally, the more powerful is the resulting method. There are many lossless data compressors applicable to texts, for example, WinRAR, WinZip, 7-zip, PeaZip. A data compressor can fail to spot any regularities within a file to be compressed in case its size is too small, as well as in the case the alphabet is too large relative to the volume. The level of compression of such a file will be greater than or equal to 1: The auxiliary data added to the compressed file by the compressor make the file larger, which may result in the compressed file being larger than the original. It is worth noting that using a weak data compressor, that is, one that spots fewer regularities, results in a lower power of the test. It means that in such a case, *H*_0_ can be chosen where *H*_1_ should have been; however, the opposite probably cannot happen with a probability higher than the pre-specified level, no matter how bad the data compressor is. In this study, similar to Reference [18], we applied the open-source data compressor 7-zip v. 18.05 (64-bit), which uses the method of data compression called Bzip2, (compressed file format .bz2). In the previous experiments, we compared different data compression methods (algorithms), namely, LZMA, Deflate and BZip2, and found the latter to be the most efficient for this kind of biological texts. The following parameters were used in the graphical user interface (GUI) for archiving: Compression level—normal; dictionary size—100kb; number of CPU threads—6.

### 2.5. Constructing Sequences for Hypothesis Testing

Using the Observer XT and the alphabet consisting of behavioral elements, we obtained sequences of letters of the complete hunting stereotypes, such as: QWEWER, SEWEHSWEHSEWWEHR, SWWHNWWNWWW. We exported all sequences obtained into text files, each file for each of the nine species, with sequences being blank-separated in each file. Thus we obtained nine raw behavioral data files. To obtain sampled data files of equal sizes, we wrote a program which randomly chose sequences from the raw data (Figure A1, see Appendix A). We then exported them, and also blank-separated them, into new sampled data text files, 300 bytes in size each. For example, one of the sample data text files included 10 behavioral sequences (291 symbols) and 9 blanks. The number of files in the output depended on the size of the raw behavioral data file. We obtained different numbers of sample data files because the lengths and numbers of behavioral sequences and, correspondingly, the sizes of the raw behavioral data files were different for each species. We obtained 64 sample data files in sum, for all nine species, in such a way that each sequence would not be exported twice, that is, it would appear in one file only. Information on the volume of data obtained is presented in Table 1.

We used Fisher’s exact test to compare the proportions of successful and unsuccessful stereotypes in different species, as well as the fractions of different types of behavioral elements. We conducted pairwise comparisons of the number of elements per behavioral stereotype in different species using the Kruskal–Wallis *H*-test. Data are expressed as median, range and first and third quartiles.

## 3. Results

### 3.1. Hunting Activity

In our experiments, 42 out of 81 (51.9%) adult *R. norvegicus*, 21 out of 26 (80.8%) *A. agrarius*, 12 out of 19 (63.2%) *P. campbelli*, 12 out of 30 (40%) *P. sungorus*, all 8 *Al. eversmanni*, all 13 *Al. curtatus*, 18 out of 46 (39.1%) *L. gregalis*, 43 out of 53 (81.1%) *Alt. tuvinicus*, and all 11 *S. araneus* demonstrated complete hunting stereotypes that ended with killing the prey. Members of all species displayed hunting behavior typical for pursuit predators, and they used a detection and pursuit phase at least over a short distance to obtain insect prey. The proportion of ‘insect hunters’ (individuals who, in principle, displayed hunting behavior) in the laboratory groups of 8 rodent species varied from 40% in *L. gregalis* to 100% in *Al. eversmanni* and *Al. curtatus*. The narrow-headed vole *L. gregalis* turned out to be the most unsuccessful hunter. Less than half of these animals hunted, and about 70% of their hunting stereotypes ended in failure. There was no evidence of learning between the first and the third trials in any species. We assume that the ratio of animals in our laboratory groups exhibiting and not exhibiting hunting behaviors reflects this ratio in natural populations.

We evaluated the success of hunting as the ratio of the number of stereotypes ending in catching prey and unsuccessful ones. According to the results of all tests, the most unsuccessful hunters turned out to be *L. gregalis*. The differences in all cases are significant, except for *Al. eversmanni*, with which *L. gregalis* did not significantly differ in this parameter (Figure 2).

Nearly all behavioral elements of hunting stereotypes turned out to be common for all the nine species, with a few differences. As distinct from rodent species, *S. araneus* never seized the prey with their paws but only with their teeth, so the elements “E” and “R” were absent in its hunting stereotype. The elements “M” and “N” (pinning the prey down to the ground with one or two paws) were observed only in shrews, and not in rodents. The ratio of the different types of behavioral elements in the hunting stereotypes of the species studied is shown in Figure 3. In *A. agrarius* and *L. gregalis*, the proportions of key elements were higher, and the proportions of auxiliary ones were lower than in all other species. The stereotypes of *Al. curtatus* and *Al. eversmanni* were significantly different from other species in that they had fewer key elements and more auxiliary ones.

To get an idea of how different species of small mammals attack prey and how they manipulate it, we compared the number of behavioral elements associated with the attack and the processing of the prey per stereotype: Bite (W), capturing the prey by paws (E), handling (R), and nibbling insects’ legs (H). The way to attack the prey differed in different species. Four hamster species belonging to the genera *Phodopus* and *Allocricetulus* start attacks from seizing an insect with paws, and do so in about 20–25% of cases. Other rodents start attacks from biting without grasping the prey with paws, and after that, they seize and handle an insect by paws. Only rarely did they grasp the cockroach with paws before biting. *S. araneus* never catch the prey with its paws, but only with the teeth, and it also can pin the insect down to the ground with the right or the left paw.

The Kruskal–Wallis criterion shows the presence of significant differences within all samples of behavioral elements: W (*H* = 287.8, *p* < 0.0001), E (*H* = 234.9, *p* = 0.0001), R (*H* = 322.9, *p* = 0.0001), H (*H* = 203.7, *p* = 0.0001). The highest number of “bites” per stereotype was most typical for *A. agrarius*, *L. gregalis*, and *S. araneus*. The hamster *P. campbelli* more often than other species studied committed acts of nibbling insects’ legs. *R. norvegicus*, *Al. curtatus*, and *Al. eversmanni*, more often than other species studied handled the prey with paws (see details in Figure 4).

### 3.2. Complexities of Hunting Stereotypes

As noted before, the compression ratios of sequences can be considered characteristic of the complexity of the files being compressed. As can be seen from Figure 5, the compression ratio of sampled data files in *R. norvegicus* (the average value 0.553) differs significantly from all other species (*U*-test, *p* < 0.01), and thus has the highest complexity. The compression ratios in other species turned out to be similar and close to the ‘insectivorous standard’, that is, to *S. araneus*. The average value of the compression ratios were 0.496 in *A. agrarius*, 0.505 in *P. campbelli*, 0.520 in *P. sungorus*, 0.514 in *Al. eversmanni*, 0.498 in *Al. curtatus*, 0.510 in *L. gregalis*, 0.495 in *Alt. tuvinicus*, and 0.513 in *S. araneus*. The compression ratios in *Al. eversmanni* (*U_emp_* = 0.5, *U_cr_* = 2, *p* = 0.041) and in *P. sungorus* (*U_emp_* = 0.0, *U_cr_* = 2, *p* = 0.019) are higher than in *Alt. tuvinicus*, and in *P. sungorus* the compression ratio is higher than in *Al. curtatus* (*U_emp_* = 7, *U_cr_* = 9, *p* = 0.045).

### 3.3. Comparison of Hunting Behaviors of Adults and Young Rats

Since the hunting stereotype in rats differs from all other species by its level of complexity and, correspondingly, by its flexibility, one can assume that learning plays a particular role in the formation of hunting behavior in this species. Earlier we applied the data compressor method to distinguish between innate and learned patterns within hunting behavior of ants and found that it is possible to extract “basic” (utterly innate) behavioral patterns by comparing behavioral sequences of different levels of complexity and flexibility [16]. In that case, the complete successful hunting stereotypes in individuals that expressed the innate pattern “all at once” turned out to be less complex than in those ones that applied the “trial and error” method. Here we compared hunting stereotypes in adult and young rats. In our experiments, 42 out of 81 (51.9%) adult and 46 out of 52 (88.5%) young rats demonstrated complete hunting stereotypes that ended with killing the prey. The sizes of the resulting text files (as described above) were 2867 bytes for adults and 4302 for young rats.

The characteristics of hunting behavior differed in adults and young rats. In the hunting stereotypes of young rats, the proportion of noise elements − 6.8% (286 of 4181) (Fisher’s exact test, *p* < 0.05), and auxiliary ones − 17.3% (725 of 4181) (*p* < 0.01) were significantly lower, and the proportion of key elements − 75.8% (3173 of 4181) were significantly higher (*p* < 0.01) than in adult rats. As can be seen from Figure 6, one hunting stereotype in young rats accounted for more “bites” (W) (*H* = 34.83, *p* < 0.01), “capturing the prey with paws” elements (E) (*H* = 39.1, *p* < 0.01), “handling” ones (R) (*H* = 5.4, *p* < 0.05), and “nibbling insects’ legs” elements (H) (*H* = 5.1, *p* < 0.05) than in adult rats.

We compared sampled data files and found that the compression ratios in adults (the average value 0.553) and young rats (0.543) were similar (*U*-test, *U_emp_* = 28.5; *U_cr_* = 13, NS).

Both in adult and young rats, elements in the hunting stereotypes are less ordered than in other species investigated. Rats often lose the prey in the process of hunting, but they continue to pursue it, which produces noise elements and makes the appearance of each next element of the behavior less predictable. Young rats do not catch prey the first time; therefore, in their stereotypes more often than in adults, such elements as bites, seizing prey with paws, and prey handling are repeated. We assume that the numerous repetitions of the “bite—seizing prey with paws” sequence in the stereotypes of young rats are due to the process of optimizing their prey-catching skills. Bites and grapples become more accurate with age in rats, even though both adult and young animals in our experiments had no prior experience with the prey. It would seem that having more key elements and less noise ones in the stereotypes of young rats should lead to a decrease in their complexity, like it was demonstrated in “basic” ants’ stereotypes [16,17]. However, the levels of complexity do not differ in young and adult rats, and rather far from the insectivorous “standard”.

## 4. Discussion and Conclusions

In our experiments, eight Muroidea species with different types of diet, mainly herbivorous, displayed skillful attacks towards the prey typical for pursuit predators. Among rodents, two species of the grasshopper mouse *Onychomys leucogaster* and *O. torridus* are known as obligate predators [19,20]. In recent years, these species have been shown to possess morphological and physiological adaptations to the interaction with relatively large, mobile and even with poisonous prey (scorpions), which allows one to consider them as specialized predators [21,22,23]. It is known that the primitive adaptation of the rodent mandibulo-dental apparatus was for an omnivorous diet rather than a herbivorous one, and that, indeed, the versatility of the feeding adaptation was the primary factor in the highly successful adaptive radiation in the order [24]. Morphological and physiological traits of Muroidea suggest the possibility of switching between plant and animal food [25]. One can assume that hunting behavior exists in rodents that lack morphological adaptations.

The specific behavioral adaptations to the insectivorous lifestyle in seven rodent species have only been discovered in our experiments. Similar results have been obtained on the bank vole *Myodes glareolus*, an omnivorous species, which demonstrated an unexpected hunting potential while lacking the morphological adaptations [26,27]. However, our study is the first case of a detailed description of hunting behavior and the analysis of the hunting stereotypes of many rodent species in comparison with a specialized insectivorous one. In order to find differences in hunting stereotypes of rodents and the insectivorous “standard”, we used the data-compression method [16] as an easy tool for comparative analysis of behaviors between and within species and groups of individuals. This method allows one to spot regularities that are difficult to detect otherwise and that can influence the complexity of behavioral sequences and then to proceed with searching for explanations of similarities and differences in behaviors.

Comparing the hunting behavior of 8 rodent species, we revealed that nearly all elements of hunting behavior were common in them, while their ratios in the behavioral sequences significantly differed. There are no differences between the levels of complexity of hunting stereotypes in such a specialized insectivorous predator as *S. araneus* and the seven rodent species, including four omnivorous hamster species, two herbivorous vole species, and the granivorous striped field mouse. The lack of differences depended neither on the frequency of the manifestation of the hunting stereotype in laboratory groups of different species nor on the ratio of the number of successful and unsuccessful stereotypes. The proportion of “insect hunters” varies widely in different species; however, the levels of complexity of hunting stereotypes in all of them are similar to the insectivorous “standard”. In sum, similar to the specialized common shrew, seven rodent species with different types of diets displayed the highly predictable hunting stereotypes, in which it was easy for the data compressor to find regularities. One can conclude that hunting behavior is highly stereotypic in the seven rodent species investigated and the insectivorous one.

In our study, the single generalist among the nine species compared, that is, *R. norvegicus,* demonstrated the highest level of complexity of hunting behavior. Comparing the hunting behavior of rats and other rodents, we believe that the specificity of stereotypes, and not the size of animals, causes the differences in their complexities. Despite the differences in size between rats and other rodents, we consider their hunting stereotypes quite comparable for two reasons. First, the differences between the ratios of the length of the bodies of a hunter and a prey range from 12% in adult rats to 15% in two *Allocricetulus* hamsters, 23% in four other rodent species, and 30% in two *Phodopus* hamsters and the shrew. It is likely from these relationships that relative sizes are not the cause of differences in hunting behavior. Second, the young rats are rather similar to at least four rodent species investigated in size, and the organizational complexity of behavioral patterns does not differ in puppies and adult rats. The high level of complexity of hunting behavior in rats reflects the high variability of their reactions to live prey, changeable manipulation of the prey, and less predictable transitions between stereotype elements. However, in the flexible and poorly predictable stereotype of the hunting behavior of *R. norvegicus,* there is a peculiar order which is manifested in the young rats. The patterns of muscular activity are improved with age without any experience, and only in minor details. The levels of complexity of hunting stereotypes do not differ in young and adult rats. In sum, one can assume that it is not learning, but the specificity of the organization of the stereotype that is responsible for the nature of the hunting behavior in rats. We speculate that rodents have different types of hunting behaviors, one of which is based on a strict and succinct insectivorous standard, and the other, perhaps characteristic of generalists, which is rather flexible, that is, less ordered and characterized by less predictable transitions between elements. We need to study more types of generalists, such as *R. norvegicus*, to confirm their different types of hunting behaviors.

In sum, we suggest that the data-compression method can be used as a universal tool to capture regularities within ethograms as “biological texts”, and this approach to behavioral analyses may well be more broadly applicable.

## Figures and Tables

**Figure 1 entropy-21-00368-f001:**
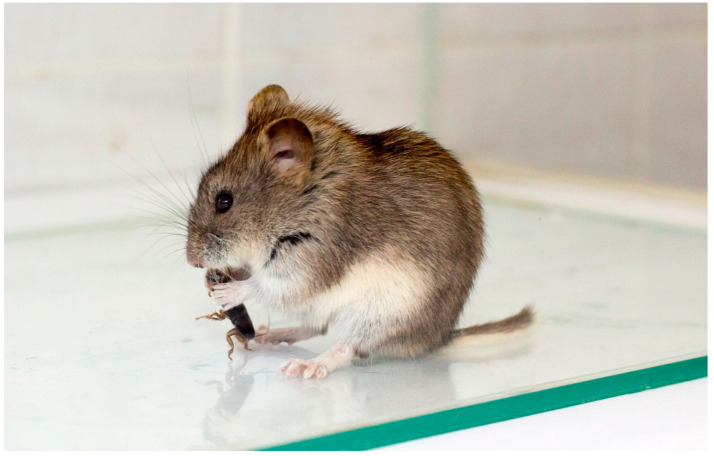
*Alticola tuvinicus* is handling the cockroach with its paws. Photo made by Galina Azarkina.

**Figure 2 entropy-21-00368-f002:**
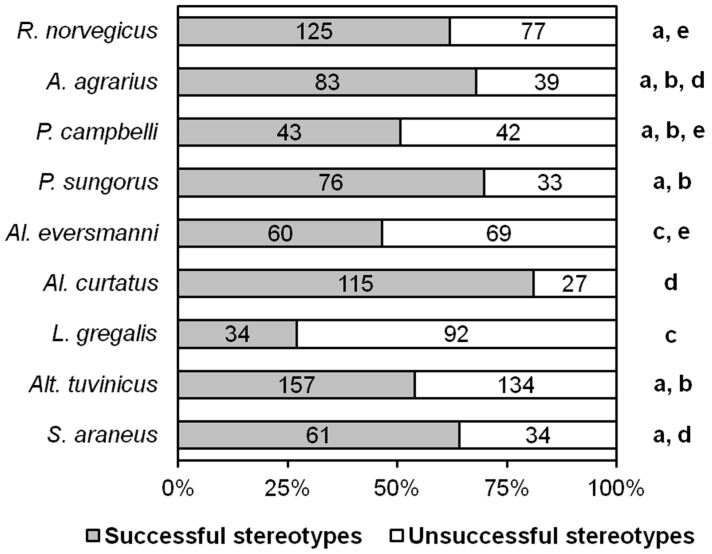
The ratios of successful and unsuccessful stereotypes of different species. Numbers indicate the number of stereotypes. The same letters (a, b, c, d, e) indicate no significant difference between species (*p* < 0.0014) according to Fisher’s exact test with Bonferroni amendment. Note that the number of successful stereotypes here is the same as within the raw data text files in Table 1.

**Figure 3 entropy-21-00368-f003:**
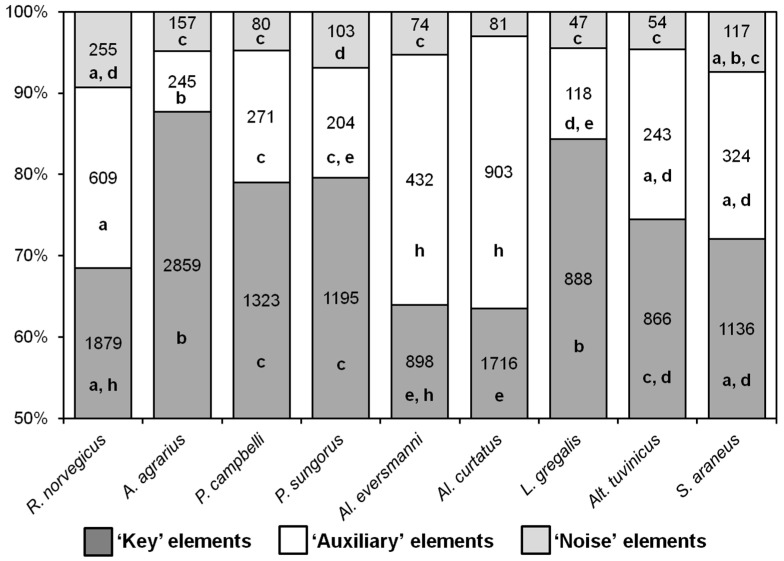
The proportion of different groups of behavioral elements in different species. Numbers indicate the number of elements. The same letters (a, b, c, d, e, h) indicate no significant difference between the same types of behavioral elements (‘Key’, ‘Auxiliary’, ‘Noise’) in different species (*p* < 0.0014) according to Fisher’s exact test with Bonferroni amendment.

**Figure 4 entropy-21-00368-f004:**
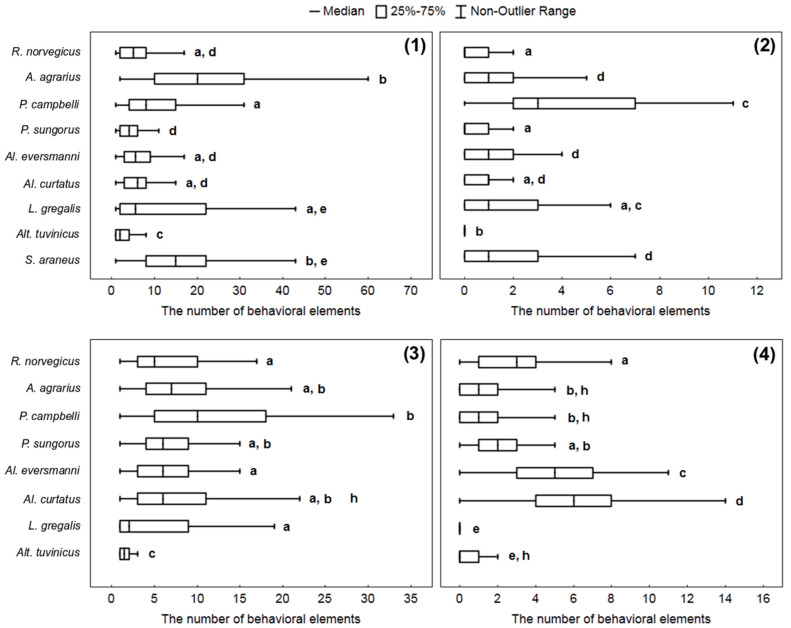
The number of elements per stereotype: (**1**) Bites “W”, (**2**) nibbling the insect’s legs “H”, (**3**) seizing an insect with paws “E”, (**4**) prey handling “R”. The same letters (a, b, c, d, e, h) indicate no significant difference between behavioral elements in different species (*p* < 0.0003) according to *H*-test with Bonferroni amendment.

**Figure 5 entropy-21-00368-f005:**
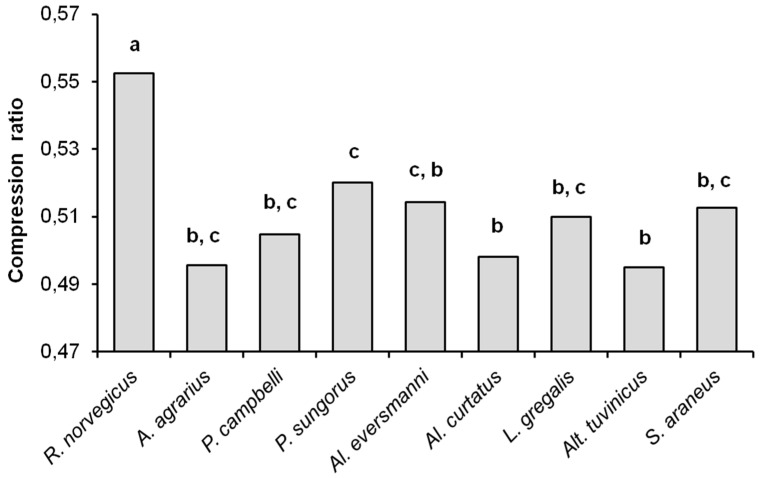
Differences between the average compression ratios of behavioral sequences in nine species. The same letters (a, b, c) indicate no significant differences between the average values of compression ratios in different species (*p* < 0.01) according to *U*-test.

**Figure 6 entropy-21-00368-f006:**
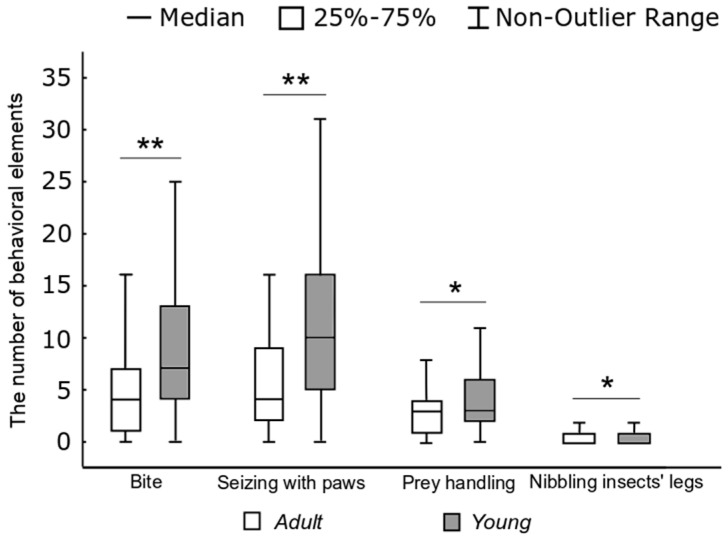
The number of behavioral elements in one hunting stereotype of adults and young rats (*H*-test, ** *p* < 0.01, * *p* < 0.05).

**Table 1 entropy-21-00368-t001:** The volumes of data obtained.

Species		Sizes of Raw Data Text Files (bytes)	Numbers of Sequences in Raw Data Text Files	Number of Sample Data Files Obtained
*R. norvegicus*	Adult	2867	125	9
	Young	4302	119	10
*A. agrarius*		3343	83	10
*P. campbelli*		1715	43	5
*P. sungorus*		1585	76	5
*Al. eversmanni*		1463	60	4
*Al. curtatus*		2814	115	9
*L. gregalis*		1086	34	3
*Alt. tuvinicus*		1319	157	4
*S. araneus*		1637	61	5

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
