# Peer review of "Using the Data-Compression Method for Studying Hunting Behavior in Small Mammals"

_entropy, 2019, doi:10.3390/e21040368_

Round 1
Reviewer 1 Report
p.p1 {margin: 0.0px 0.0px 0.0px 0.0px; font: 12.0px Arial} p.p2 {margin: 0.0px 0.0px 0.0px 0.0px; font: 12.0px Arial; min-height: 14.0px}Reznikova and colleagues placed prey insects into the home cages of individuals from a range of small mammalian species, and for each encounter generated a sequence of symbols, where each symbol corresponds to a distinct behavioural event such as biting or jumping. They then applied a simple data compression algorithm, Bzip2, to reduce the size of text files containing examples of the sequences selected at random to give an initial file size of equal length. Compression reduced the size of files containing ethograms for most species to a similar extent, but for Norway rats the compression was significantly less, suggesting a higher complexity to the sequences used to describe rat prey-capture. The authors convincingly rule out several possible alternative explanations for the differences by reporting on the differences in the number of attacks made, and the relative frequency of each event type, across species. They also report on differences between old and young rats. Together the data support the conclusion that rats have a more unpredictable behavioural repertoire, which changes with age/experience.
Overall, I enjoyed reading this article, and found it to be a neat demonstration of an interesting approach to behavioural analyses that may well be more broadly applicable. There are some grammatical and typographical issues (identified below), but overall the paper is well written, and the figures are clear and produced to a good standard. The statistical tests were chosen appropriately and reported clearly. I have two general points for consideration.
First, as the authors point out on page 5, ‘The possibility to compress information realised by different data compressors is highly dependent on the chosen method of compression, that is, on the algorithm used to find regularities in the file to be compressed’. The authors list a number of example algorithms and state that they chose to use the Bzip2 algorithm, but a justification of this particular choice is missing from the text, as are details of how this algorithm works, and both should be included in a revision of the manuscript. A quick online search suggests that at the core of this particular algorithm is a method called block-sorting (Burrows-Wheeler transform), which essentially truncates long sequences in which a symbol is repeated numerous times. Hence I would expect the number of times that a symbol (in this case a behaviour) is repeated without interruption (by a different behaviour) to have a large influence on the data compression. It is not clear from the ‘data encoding’ section how repeated behavioural events are represented. For example, would five successive bites followed by two jumps be represented as WWWWWJJ or simply as WJ? If the latter, as I suspect, then truncation will have no effect on the results, but this is not currently clear from the text. In either case, the second component of the Bzip2 algorithm, i.e., Huffman coding on blocks containing 4 symbols, may be relevant - perhaps rat attack sequences are equally predictable but differ in terms of the typical length of repeated sequences, as may be picked up by an algorithm that exploited predictability by considering blocks of a different length? The authors might consider using another compression algorithm for comparison, or basing the analysis on the minimal file-size generated after applying two or more compression algorithms to each file (though a more nuanced discussion of how the Bzip algorithm exploits regularities to reduce file size would suffice).
Second, can the authors provide a brief justification for looking only at data pertaining to successful attacks. I wonder whether differences in the compression ratio between species may be a reflection of differences in the aspects of a sequence that make it successful (result in eating) for that species, rather than to the behavioural strategy typically adopted by that species per se. Factors that could conceivably differentiate species in terms of the conditions that lead to the recording of a successful attack, rather than the behavioural strategy itself, may be e.g., relative size of the prey or morphology of the mammal. Consider that a rat may use the same strategy (behavioural sequence) to the other species, but if morphological/motivational/perceptual differences etc. make that same strategy less successful for the rat than in other species, then the data recorded and analysed for rats may be systematically different (i.e., less predictable). I suspect that this is not an issue for the current results, but some justification for concentrating only on successful events I think is warranted.
Note also that the figures identify the species by their formal names, e.g., ‘R norvegicus’ but the text throughout refers to their non-formal names e.g., ‘Norway rat’. It would improve the readability if in the text the formal name appeared in parentheses after each use of the informal name, so that the reader unfamiliar with the translation can readily compare between the text and figures.
Typos (line numbers shown):
46 - “based on the modelling” - remove the
49 - “approximately adequate” - odd, revise
55 - “issues such the search” - add ‘as’
61 - “all kind of“ -‘kinds’
83 - “researches” - revise to “research”
86 - “without resorting to rearing naive animals” - Suggest removing this.
88 - “appeared” - change to ‘appears’
111 - “52 young naive rats” - naive to what?
122 - “victims” - 'prey'?
133 - “three exemplars of cockroach in turn” - I think you mean ‘three cockroaches in turn’
205 - “each file for each of the three species“ - do you mean ‘one file for each of the prey’?
301 - “one” - typo somewhere here - unclear
333 - “neither” - delete this.
335 - “The similarity…” - I cannot parse this sentence - revise
341 - “If we may say so…” - unclear sentnce - I suggest removing.
360 - “neither” - delete this
Author Response
Request
First, as the authors point out on page 5, ‘The possibility to compress information realised by different data compressors is highly dependent on the chosen method of compression, that is, on the algorithm used to find regularities in the file to be compressed’. The authors list a number of example algorithms and state that they chose to use the Bzip2 algorithm, but a justification of this particular choice is missing from the text, as are details of how this algorithm works, and both should be included in a revision of the manuscript. A quick online search suggests that at the core of this particular algorithm is a method called block-sorting (Burrows-Wheeler transform), which essentially truncates long sequences in which a symbol is repeated numerous times.
Hence I would expect the number of times that a symbol (in this case a behaviour) is repeated without interruption (by a different behaviour) to have a large influence on the data compression. It is not clear from the ‘data encoding’ section how repeated behavioural events are represented. For example, would five successive bites followed by two jumps be represented as WWWWWJJ or simply as WJ? If the latter, as I suspect, then truncation will have no effect on the results, but this is not currently clear from the text. In either case, the second component of the Bzip2 algorithm, i.e., Huffman coding on blocks containing 4 symbols, may be relevant - perhaps rat attack sequences are equally predictable but differ in terms of the typical length of repeated sequences, as may be picked up by an algorithm that exploited predictability by considering blocks of a different length? The authors might consider using another compression algorithm for comparison, or basing the analysis on the minimal file-size generated after applying two or more compression algorithms to each file (though a more nuanced discussion of how the Bzip algorithm exploits regularities to reduce file size would suffice).
Answer.
To meet this concern, we, first, replaced “information” with the word “sequence”. Then, we added the following fragment (lines 184-188): In this study, similar to [26] we applied the open-source data compressor 7-zip v. 18.05 (64-bit), which uses the method of data compression called Bzip2, (compressed file format .bz2). In the previous experiments, we compared different data compression methods (algorithms), namely, LZMA, Deflate and BZip2, and found the latter to be the most efficient on this kind of biological texts.
We also clarified how repeated behavioral elements were represented (lines 141-145): Video recordings were processed by assigning a letter to each of the elements of behavior, in the order of their appearance, without taking into account their duration. This results in a sequence of letters. For example, if an animal stalks a prey for 10 seconds and then bites, the sequence would be QW. If an animal committed a behavioral act several times in a row, we recorded this as follows: one bite – W, three bites – WWW, a bite and two captures with paws – WEE.
Request:
Second, can the authors provide a brief justification for looking only at data pertaining to successful attacks. I wonder whether differences in the compression ratio between species may be a reflection of differences in the aspects of a sequence that make it successful (result in eating) for that species, rather than to the behavioural strategy typically adopted by that species per se. Factors that could conceivably differentiate species in terms of the conditions that lead to the recording of a successful attack, rather than the behavioural strategy itself, may be e.g., relative size of the prey or morphology of the mammal. Consider that a rat may use the same strategy (behavioural sequence) to the other species, but if morphological/motivational/perceptual differences etc. make that same strategy less successful for the rat than in other species, then the data recorded and analysed for rats may be systematically different (i.e., less predictable). I suspect that this is not an issue for the current results, but some justification for concentrating only on successful events I think is warranted.
Answer.
The question of analyzing unsuccessful stereotypes is fascinating, and we are now dealing with it in a separate study.
To meet this concern in general, we added the following fragment in Discussion (lines 359-367): Comparing the hunting behavior of rats and other rodents, we believe that the specificity of stereotypes, and not the size of animals, causes the differences in their complexities. Despite the difference in size between rats and other rodents, we consider their hunting stereotypes quite comparable for two reasons. First, the differences between the ratios of the length of the bodies of a hunter and a prey range from 12% in adult rats to 15% in two Allocricetulus hamsters, 23% in four other rodent species, and 30% in two Phodopus hamsters and the shrew. It is likely from these relationships that relative sizes are not the cause of differences in hunting behavior. Second, the young rats are rather similar to at least four rodent species investigated in size, and the organizational complexity of behavioral patterns does not differ in puppies and adult rats.
Request:
Note also that the figures identify the species by their formal names, e.g., ‘R norvegicus’ but the text throughout refers to their non-formal names e.g., ‘Norway rat’. It would improve the readability if in the text the formal name appeared in parentheses after each use of the informal name, so that the reader unfamiliar with the translation can readily compare between the text and figures.
Answer:
Fixed.
Typos (line numbers shown):
Answer:Fixed
46 - “based on the modelling” - remove the
49 - “approximately adequate” - odd, revise
55 - “issues such the search” - add ‘as’
61 - “all kind of“ -‘kinds’
83 - “researches” - revise to “research”
86 - “without resorting to rearing naive animals” - Suggest removing this.
88 - “appeared” - change to ‘appears’
111 - “52 young naive rats” - naive to what?
122 - “victims” - 'prey'?
133 - “three exemplars of cockroach in turn” - I think you mean ‘three cockroaches in turn’
Answer: done
205 - “each file for each of the three species“ - do you mean ‘one file for each of the prey’?
Answer: typo fixed: nine species
301 - “one” - typo somewhere here - unclear
333 - “neither” - delete this.
335 - “The similarity…” - I cannot parse this sentence – revise
341 - “If we may say so…” - unclear sentnce - I suggest removing.
360 - “neither” - delete this

Reviewer 2 Report
The authors present work that shows how hunting behaviors can be discretized into ethograms and compressed using open source compression software to indicate relative levels of behavioral complexity. The authors add behavioral data from six new species to their previously published dataset (Reznikova et al. 2017) and add an ontogenetic comparison for one species. The authors claim that this method may be capable of using behavioral data to gauge hunting potential in species with morphologies that are not specialized for hunting.
The manuscript does not currently meet the MDPI ethical standards (https://www.mdpi.com/reviewers) in the following ways:
· “Manuscripts should only report results that have not been submitted or published before, even in part.”
o Data from three of the six species was previously published (Reznikova et al. 2017)
· “Manuscripts must be original and should not reuse text from another source without appropriate citation.”
o Approximately one third of the text of this paper is copied word-for-word from a previous publication by the authors (Reznikova et al. 2017)
· “For biological studies, the studies reported should have been carried out in accordance with generally accepted ethical research standards.”
o Did the authors seek and gain approval from an animal welfare and/or ethics board for their experimental protocols? If so, please list them either in the animal section of the methods, or in their own subsection after the conclusion. Unapproved experimentation on animals will be recommended for rejection.
Major comments:
· Entropy-related
o Overall: The compression method does not actually identify strings of repeated behaviors for biological relevance. It only implies the presence of these regularities/stereotypes through the compression ratio. Furthermore, this method cannot determine whether different species have similar behavioral strategies for hunting, only the relative complexity. These distinctions are worth noting in the text.
o Lines 59-61 – How does this method actually work? There are several vague and repetitive descriptions of the method that do not actually describe how the method works. This would be a good place to introduce the idea of a compression ratio.
o 170-173 – While the equations are defined here, the length (t) of fragments is not stated. Was t the same for all species? How was t determined?
o Lines 177-179 –Since it is impossible to prove that a lower bound on K entropy can be determined, is it provable that the relative compressed sizes of the files reflects the true relationship between their respective complexities?
o Lines 211-215 – Why are there different numbers of files for each species? Should an equal length of sampled data be used for adequate comparison? How can a file have 291 symbols that represent 10 behavioral sequences, if all behavioral sequences must be of the same size?
· Biological
o Since the methods and a subset of the data have previously been published, what distinguishes this new paper? Why were additional species included? In the discussion, the authors state that the method could be used to identify behavioral hunting potential in animals that lack morphological adaptations. This concept would be useful in the introduction. Furthermore, the ontogenetic comparison of Norway rat hunting behavior is unmotivated. Why are the authors performing this experiment? Both of these ideas should be thoroughly discussed in the introduction.
o All species names should be italicised. Also, all studied species should have both a common name and a species name listed in the text. Currently the common name is used in the manuscript text, whereas the species names are used in the figures. This is very confusing, especially when many of the species names are not identified in the text. A practical solution would be to use the common names only once when introducing each species here and use the species names throughout the rest of the manuscript.
o Line 73 – This paragraph motivates the study by describing an open question in animal behavior. It would work well as the first paragraph of the Introduction.
o Line 76 – Variability is not a mechanism of speciation.
o Line 109-124 – Why were these species selected? Also, it should be noted that the adult Norway rat, striped field mouse, and common shrew data have been previously published. Also, why were juvenile rats included, when all others were adults?
o Line 133 – Since the authors state that the subjects had not previously encountered live insects, was there evidence of any learning between the first and the third trials?
o Lines 142-153 – Were behaviors identified only when a behavior changed, rather than at regular intervals (as in classical ethograms)? For example, can an ethogram have two bites in a row? Is a series of small nibbles considered one behavior or several?
o Line 320 – The idea that hunting potential exists in behavioral data despite lacking morphological specializations for hunting is a key concept that should be introduced in the introduction. However, shouldn't this be expected in omnivorous species? Mammalian specialization from omnivory to herbivory is associated with multiple morphological changes, whereas the omnivorous morphology is generally considered capable of hunting (Eubanks, 2005 “Predaceous herbivores and herbivorous predators: The biology of omnivores and the ecology of omnivore-prey interactions,” in Ecology of Predator-Prey Interactions; Stuart O. Landry, "The Rodentia as Omnivores," The Quarterly Review of Biology 45, no. 4 (Dec., 1970): 351-372.)
o Line 340 – “Somewhere deep inside the narrow-headed vole is a shrew” is an unsupportable statement. If these species were closely related, there would be an argument for lineage-specific effects. Rather, the distant relationship between these species implies behavioral convergence, which may or may not be due to similar selective pressures.
o Line 350 - While a peculiar order is found for the Norway rat, the order of behaviors is not described for other species. Furthermore, since the sequence of “bite” and “grasp prey with paws” is repeated, is it possible that these should be considered the same behavior? Do bites ever happen without grasping the prey with paws? What happens if these events cooccur in time?
· Figures
o The way that significance is indicated in all of the figures is very confusing. Perhaps a table of p values for pairwise comparisons would be useful? At the minimum, the caption should include more information regarding the meaning of the letters, perhaps with an example.
Minor Comments:
· Biological:
o Line 39 – Instead of “texts,” which is a very broad term, the concept a range of capable behaviors is often called a “repertoire” or “dictionary” of behaviors. (Drummond, H. (1981). The nature and description of behavioral patterns. In P. P. G. Bateson (Ed.), Perspectives in ethology (Vol. 4, pp. 1–73). New York: Plenum Press.)
o Line 126 – How were the lobster cockroaches obtained and cared for?
o Line 130 – Why were different frame rates used, and are the data from the two different framerates of video comparable?
o According to Babinska-Werka (1981, Acta Thereologica), the striped field mouse has a large proportion of insects in its diet, so it is not necessarily correct to assume that this mouse is strictly granivorous and lacks any hunting skills
o Line 133 – How can the authors be certain that wild-caught insectivores had never encountered live insects before?
· Writing, word choice, grammar
o Title: The title is misleading because the behavioral stereotypes themselves are not being identified or studied by this compression method.
o Throughout – Change “victim” to “prey”
o Abstract – The work is not motivated at all, and the comparison between juvenile and adult rats is not explained.
o Lines 171, 205, 207, 208, 209 – Using the label "resulting" for a file created in an intermediate step is both confusing and misleading, especially when the word "resulting" is used as an adjective throughout the manuscript. It may be useful to use more precise language for each step in the process. Perhaps terms like "raw behavioral data," sampled data," and "compressed file" would be more informative.
o Line 15 – Reword for clarity
o Line 17 – The terms “focused,” “succinct,” “flexibility,” and “stereotype elements” are all terms with precise meaning that should be defined, both with respect to information theory and with respect to biological relevance.
o Line 32 – What are “signal activities?” There are several other papers using the entropy to analyze many aspects of animal behavior, including
§ Stress response in tuna: https://doi.org/10.1371/journal.pone.0028241
§ Predator evasion: https://doi.org/10.1038/s41467-017-00373-2
§ Division of labor: https://doi.org/10.1007/s00040-007-0923-z
o Line 43 – What are “regularities?” This term should be precisely defined at its first use in the text. An example of how regularities in behavior are biologically relevant would be helpful to justify this method.
o Line 51 – This example of genetic data is not necessary
o Line 67 – What is meant by the phrase “reason precisely?”
o Line 68 – This is the first time that a “difference in complexity” is stated as the method of comparison. Please introduce and explain this earlier.
o Line 82 – Rewording for clarity is suggested here.
o Line 87 – Animal kingdom doesn’t need to be capitalized
o Lines 99-106 – These are results that should not be in the introduction
o Lines 109-124 – The description of how each animal was captured was confusing. I would recommend stating the capture method as the species was introduced. For example, " we used 81 (41 females and 40 males) lab-reared non-pedigree adult Norway Rats (Rattus norvegicus),” etc
o Line 122 – How are reactions “spontaneous?”
o Line 133 – The phrase “exemplars of” is not needed here
o Line 137 – This definition of “stereotype” should be moved to the introduction
o Line 140 – Change “video records” to “video recordings”
o Line 158 – “an assessment of certain characteristics of a text” is vague. For which characteristics are you searching?
o Line 163 – To which probabilistic approach are the authors referring?
o Line 179-184 – Move the section “So we can use the compression ratio… …is the resulting method” to the introduction
o Line 193 – “one that cannot spot fewer regularities” is difficult to parse due to double negatives. Suggest rewording for clarity.
o Lines 198-201 – Move these numbers to the results.
o Line 204 – This line refers to three species, whereas nine species were considered in this study.
o Line 206 – Please provide more details regarding the “custom program.”
o Line 223 – The manuscript would benefit from separate subsections for the results of comparison across adult rat species and the ontogenetic comparison within rats
o Line 316 – The comparison to the grasshopper mouse should be motivated and more thoroughly described.
o Lines 335-339 – “The proportion of… …ended in failure” should be moved to results
Lines 358-368 –All of the conclusion is present, nearly word for word, in the discussion. The manuscript does not benefit from this repetition.

Author Response
Requests:
The manuscript does not currently meet the MDPI ethical standards (https://www.mdpi.com/reviewers) in the following ways:
“Manuscripts should only report results that have not been submitted or published before, even in part.”
Data from three of the six species was previously published (Reznikova et al. 2017)
Answer:
Not three out of six, but 3 out of 9, that is, new are data on six species, which is essential.
“Manuscripts must be original and should not reuse text from another source without appropriate citation.”
Approximately one third of the text of this paper is copied word-for-word from a previous publication by the authors (Reznikova et al. 2017)
Answer:
We removed auto-citations
“For biological studies, the studies reported should have been carried out in accordance with generally accepted ethical research standards.”
Did the authors seek and gain approval from an animal welfare and/or ethics board for their experimental protocols? If so, please list them either in the animal section of the methods, or in their own subsection after the conclusion. Unapproved experimentation on animals will be recommended for rejection.
Answer:
We added the following fragment (lines 110-113): All experiments with rodents were performed in accordance with the rules adopted by the European Convention for the Protection of Vertebrate Animals used for Experimental and other Scientific Purposes. The experimental protocol was approved by the Bioethical Committee of Severtsov Institute of Ecology and Evolution Russian Academy of Sciences (protocol 22, 3.05.2018).
Major comments:
Entropy-related
Overall: The compression method does not actually identify strings of repeated behaviors for biological relevance. It only implies the presence of these regularities/stereotypes through the compression ratio. Furthermore, this method cannot determine whether different species have similar behavioral strategies for hunting, only the relative complexity. These distinctions are worth noting in the text.
Answer.
To meet this concern, we added the following fragment (lines 52-63): The data-compression method suggested by Ryabko et al. [30] is based on the concept of Kolmogorov complexity and allows to search for regularities within sequences of symbols using data compressors (see details in: [26,27,30]). The main idea behind the data compression method is that it is able to capture all kind of regularities in a “text”, and do so in a way amenable to formal statistical analysis. By regularity we mean any characteristic of a text that makes it more predictable, such as repeating subsequences, letters or sequences that are more likely to occur when preceded or followed by certain other letters or subsequences, and so on. In general, regularities may be complex and involve arbitrary computable functions, but when applying a data compressor we restrict ourselves to those regularities it can capture. When comparing ethograms of different species as biological “texts”, the method implies the presence of regularities within behavioral patterns and finds all differences in the complexity of the ethograms. If we see such differences, then additional biological data are needed to explain them.
Request
Lines 59-61 – How does this method actually work? There are several vague and repetitive descriptions of the method that do not actually describe how the method works. This would be a good place to introduce the idea of a compression ratio.
Answer:
All details of how does the method work are described in [Ryabko et al., 2013], and here we were trying to avoid auto- citations. To meet this concern, we added a following fragment (lines 184-188): In this study, similar to [26] we applied the open-source data compressor 7-zip v. 18.05 (64-bit), which uses the method of data compression called Bzip2, (compressed file format .bz2). In the previous experiments, we compared different data compression methods (algorithms), namely, LZMA, Deflate and BZip2, and found the latter to be the most efficient on this kind of biological texts.
Request:
170-173 – While the equations are defined here, the length (t) of fragments is not stated. Was t the same for all species? How was t determined?
Answer:
We added the following fragment (lines 194-203): We exported all sequences obtained into text files, each file for each of the nine species, with sequences being blank-separated in each file. Thus we obtained nine raw behavioral data files. To obtain sampled data files of equal sizes, we wrote a program which randomly chose sequences from the raw data (Figure 7, see Appendix A). We then exported them, also blank-separated, into new sampled data text files, 300 bytes in size each. For example, one of the sample data text files included 10 behavioral sequences (291 symbols) and 9 blanks. The number of files in the output depended on the size of the raw behavioral data file. We obtained different numbers of sample data files because the lengths and numbers of behavioral sequences and, correspondingly, the sizes of the raw behavioral data files were different for each species.
Request:
Lines 177-179 –Since it is impossible to prove that a lower bound on K entropy can be determined, is it provable that the relative compressed sizes of the files reflects the true relationship between their respective complexities?
Answer:
No, it is not so. That is, one sequence may shrink better than the other, and Kolmogorov complexity may be higher. But if the sequences are generated by a stationary source and universal codes are used (typically data compressors are universal codes) then the probability of such a situation tends to 0 as the sequence length grows.
Request:
Lines 211-215 – Why are there different numbers of files for each species? Should an equal length of sampled data be used for adequate comparison? How can a file have 291 symbols that represent 10 behavioral sequences, if all behavioral sequences must be of the same size?
Answer:
We added the following fragment (lines 192-205) and added Appendix A and Table 1.
Using The Observer XT and the alphabet consisting of behavioral elements, we obtained sequences of letters of the complete hunting stereotypes, such as: QWEWER, SEWEHSWEHSEWWEHR, SWWHNWWNWWW. We exported all sequences obtained into text files, each file for each of the nine species, with sequences being blank-separated in each file. Thus we obtained nine raw behavioral data files. To obtain sampled data files of equal sizes, we wrote a program which randomly chose sequences from the raw data (Figure 7, see Appendix A). We then exported them, also blank-separated, into new sampled data text files, 300 bytes in size each. For example, one of the sample data text files included 10 behavioral sequences (291 symbols) and 9 blanks. The number of files in the output depended on the size of the raw behavioral data file. We obtained different numbers of sample data files because the lengths and numbers of behavioral sequences and, correspondingly, the sizes of the raw behavioral data files were different for each species. We obtained 64 sample data files in sum, for all nine species, in such a way that each sequence would not be exported twice, that is, it would appear in one file only. Information on the volume of data obtained is presented in Table 1.
Biological
Request:
Since the methods and a subset of the data have previously been published, what distinguishes this new paper? Why were additional species included?
Answer:
Previously the method has been presented with the data on two rodent species and one insectivorous species as an example. This is the first evidence that eight Muroidea species with different types of diet, mainly herbivorous, displayed skillful attacks towards the prey typical for pursuit predators. The data-compression method has been applied here for the first time to compare the great diversity of species.
Request:
In the discussion, the authors state that the method could be used to identify behavioral hunting potential in animals that lack morphological adaptations. This concept would be useful in the introduction.
Answer:
We added the following fragment in the Discussion (lines 323-332): Among rodents, two species of the grasshopper mouse Onychomys leucogaster and O. torridus are known as obligate predators [15,16]. In recent years, these species have been shown to possess morphological and physiological adaptations to the interaction with relatively large, mobile and even with poisonous prey (scorpions), which allows one to consider them as specialized predators [28,29,35]. It is known that the primitive adaptation of the rodent mandibulo-dental apparatus was for an omnivorous diet rather than a herbivorous one, and that, indeed, the versatility of the feeding adaptation was the primary factor in the highly successful adaptive radiation in the order [14]. Morphology and physiology of omnivorous in Muroidea, suggest the possibility of switching between plant and animal food [5]. One can thus assume that behavioral hunting potential exists in rodents lack morphological adaptations.
Request:
Furthermore, the ontogenetic comparison of Norway rat hunting behavior is unmotivated. Why are the authors performing this experiment? Both of these ideas should be thoroughly discussed in the introduction.
Answer:
We added the following fragments in the Abstract: The levels of complexities of hunting stereotypes in young and adult rats are similar, so one can assume that it is not learning, but the specificity of the organization of the stereotype that is responsible for the nature of the hunting behavior in rats.
and in the Results (lines 285-292):
Since the hunting stereotype in rats differs from all other species by its level of complexity and, correspondingly, by its flexibility, one can assume that learning plays a particular role in the formation of hunting behavior in this species. Earlier we applied the data compressor method to distinguish between innate and learned patterns within hunting behavior of ants and found that it is possible to extract “basic” (utterly innate) behavioral patterns by comparing behavioral sequences of different levels of complexity and flexibility [30]. In that case, the complete successful hunting stereotypes in individuals that expressed the innate pattern “all at once” turned out to be less complex than in those ones that applied the ‘trial and error’ method.
Request:
All species names should be italicised. Also, all studied species should have both a common name and a species name listed in the text. Currently the common name is used in the manuscript text, whereas the species names are used in the figures. This is very confusing, especially when many of the species names are not identified in the text. A practical solution would be to use the common names only once when introducing each species here and use the species names throughout the rest of the manuscript.
Answer:
Fixed
Request:
Line 73 – This paragraph motivates the study by describing an open question in animal behavior. It would work well as the first paragraph of the Introduction.
Answer:
Done
Request:
Line 76 – Variability is not a mechanism of speciation.
Answer:
We removed this phrase.
Request:
Line 109-124 – Why were these species selected? Also, it should be noted that the adult Norway rat, striped field mouse, and common shrew data have been previously published.
Answer:
There is a phrase in Introduction, in which we now added several new words: “In this study, we compare, for the first time, the organizational complexity of species-specific hunting stereotypes in nine species of small mammals with various diets and different levels of phylogenetic relationship.” We tried to cover a significant variety of members of Muroidea with different types of diets. Note, that the insectivorous shrew is our constant standard revealed in the previous study (Reznikova et al., 2017), and from eight rodent species only two have been studied earlier.
Request:
Also, why were juvenile rats included, when all others were adults?
Answer:
We gave the answer above.
Request:
Line 133 – Since the authors state that the subjects had not previously encountered live insects, was there evidence of any learning between the first and the third trials?
Answer:
We added the phrase There were no evidence of learning between the first and the third trials in any species. (lines 223-224)
Request:
Lines 142-153 – Were behaviors identified only when a behavior changed, rather than at In this study, we compare, for the first time, the organizational complexity of species-specific hunting stereotypes in nine species of small mammals with various diets and different levels of phylogenetic relationship.row? Is a series of small nibbles considered one behavior or several?
Answer:
To meet this concern, we added the following fragment (lines 141-145): Video recordings were processed by assigning a letter to each of the elements of behavior, in the order of their appearance, without taking into account their duration. This results in a sequence of letters. For example, if an animal stalks a prey for 10 seconds and then bites, the sequence would be QW. If an animal committed a behavioral act several times in a row, we recorded this as follows: one bite – W, three bites – WWW, a bite and two captures with paws – WEE.
Request:
Line 320 – The idea that hunting potential exists in behavioral data despite lacking morphological specializations for hunting is a key concept that should be introduced in the introduction. However, shouldn't this be expected in omnivorous species? Mammalian specialization from omnivory to herbivory is associated with multiple morphological changes, whereas the omnivorous morphology is generally considered capable of hunting (Eubanks, 2005 “Predaceous herbivores and herbivorous predators: The biology of omnivores and the ecology of omnivore-prey interactions,” in Ecology of Predator-Prey Interactions; Stuart O. Landry, "The Rodentia as Omnivores," The Quarterly Review of Biology 45, no. 4 (Dec., 1970): 351-372.)
Answer:
To meet this concern, we added the following fragment to the Discussion (lines 322-332): Among rodents, two species of the grasshopper mouse Onychomys leucogaster and O. torridus are known as obligate predators [15,16]. In recent years, these species have been shown to possess morphological and physiological adaptations to the interaction with relatively large, mobile and even with poisonous prey (scorpions), which allows one to consider them as specialized predators [28,29,35]. It is known that the primitive adaptation of the rodent mandibulo-dental apparatus was for an omnivorous diet rather than a herbivorous one, and that, indeed, the versatility of the feeding adaptation was the primary factor in the highly successful adaptive radiation in the order [14]. Morphology and physiology of omnivorous in Muroidea, suggest the possibility of switching between plant and animal food [5]. One can thus assume that behavioral hunting potential exists in rodents lack morphological adaptations.
Request:
Line 340 – “Somewhere deep inside the narrow-headed vole is a shrew” is an unsupportable statement. If these species were closely related, there would be an argument for lineage-specific effects. Rather, the distant relationship between these species implies behavioral convergence, which may or may not be due to similar selective pressures.
Answer:
removed.
Request:
Line 350 - While a peculiar order is found for the Norway rat, the order of behaviors is not described for other species. Furthermore, since the sequence of “bite” and “grasp prey with paws” is repeated, is it possible that these should be considered the same behavior? Do bites ever happen without grasping the prey with paws? What happens if these events cooccur in time?
Answer:
We added the following fragment (lines 252-257): The way to attack the prey differed in different species. Four hamster species belonging to the genera Phodopus and Allocricetulus start attacks from seizing an insect with paws, and do so in about 20-25% of cases. Other rodents start attacks from biting without grasping the prey with paws, and after that they seize and handle an insect by paws. Only rarely did they grasp the cockroach with paws before biting. S. araneus never catch the prey with its paws, but only with the teeth, and it also can pin the insect down to the ground with the right or the left paw.
Figures
Request:
The way that significance is indicated in all of the figures is very confusing. Perhaps a table of p values for pairwise comparisons would be useful? At the minimum, the caption should include more information regarding the meaning of the letters, perhaps with an example.
Answer:
We changed all captions like this: Fig 2 (as an example): The same letters indicate no significant difference between species (p < 0.0014) according to Fisher’s exact test with Bonferroni amendment.
Minor Comments:
Biological:
Request:
Line 39 – Instead of “texts,” which is a very broad term, the concept a range of capable behaviors is often called a “repertoire” or “dictionary” of behaviors. (Drummond, H. (1981). The nature and description of behavioral patterns. In P. P. G. Bateson (Ed.), Perspectives in ethology (Vol. 4, pp. 1–73). New York: Plenum Press.)
Answer:
To meet this concern, we added the following fragments: In the field of comparative ethology, the study of the structure of behavior is based on identification of specific behavioral patterns [1,3], which, in turn, are viewed as classes defined by regularities in one or more of five domains: location, orientation, physical topography, intrinsic properties, and physical effects (Drummond, 1981). (lines 35-38).
and: We analyze ethograms as a case of “biological texts”, that is, as sequences of symbols from a finite alphabet (Ryabko et al., 2013). (lines 46-47).
Request:
Line 126 – How were the lobster cockroaches obtained and cared for?
Answer:
We added (lines 105-107): We used imago and last instar nymphs of the lobster cockroach Nauphoeta cinerea (27.93 ± 0.40 mm) as a live mobile prey. Insects were kept and bred in our laboratory in separate containers.
Request:
Line 130 – Why were different frame rates used, and are the data from the two different framerates of video comparable?
Answer: (see lines 118-121)
The rate of hunting in species varied, that is why, for the most "fast" hunters, to distinguish the behavioral elements better, we used an increased frame rate when shooting. The data obtained when shooting with 25 and 60 frames per second are comparable, taking into account differences in the speed of hunting in the studied species.
Request:
According to Babinska-Werka (1981, Acta Thereologica), the striped field mouse has a large proportion of insects in its diet, so it is not necessarily correct to assume that this mouse is strictly granivorous and lacks any hunting skills.
Answer:
As Landry (1970) noted, the feeding habits of rodents shows many species in all major lines of rodent evolution to be, to a surprising degree, carnivorous, piscivorous, or insectivorous. However, a large proportion of insects in the diet does not mean hunting skills but rather collecting dead and immobile insects. We were the first to demonstrate hunting skills in the striped field mouse.
Request:
Line 133 – How can the authors be certain that wild-caught insectivores had never encountered live insects before?
Answer:
There was an inaccuracy in Methods, now it is as follows (lines 100-105.): Out of the 26 mice, 9 (6 females and 3 males) were born in the laboratory being the progeny of the wild-caught mice, while 10 males and 7 females, as well as all of S. araneus, L. gregalis, and 4 individuals of Alt. tuvinicus were captured in the natural environment with the use of Sherman’s traps. Members of the other species were descendants caught in the wild and have not previously encountered live insects.
Writing, word choice, grammar
Title: The title is misleading because the behavioral stereotypes themselves are not being identified or studied by this compression method.
Answer:
We changed the title as follows: Using the data-compression method for studying hunting behavior in small mammals
Throughout – Change “victim” to “prey”
Answer: done
Abstract – The work is not motivated at all, and the comparison between juvenile and adult rats is not explained.
Answer: we re-wrote the abstract, in particular, changed and/or added the following phrases: Using the data-compression method we revealed a similarity between hunting behavior of the common shrew, which is insectivorous, and several rodent species with different types of diet. Seven rodent species studied displayed succinct, highly predictable hunting stereotypes, in which it was easy for the data compressor to find regularities. (lines 15-18).
The levels of complexities of hunting stereotypes in young and adult rats are similar, so one can assume that it is not learning, but the specificity of the organization of the stereotype that is responsible for the nature of the hunting behavior in rats. (lines 20-22)
We suggest that the data-compression method may well be more broadly applicable to behavioral analysis (lines 25-26).
Lines 171, 205, 207, 208, 209 – Using the label "resulting" for a file created in an intermediate step is both confusing and misleading, especially when the word "resulting" is used as an adjective throughout the manuscript. It may be useful to use more precise language for each step in the process. Perhaps terms like "raw behavioral data," sampled data," and "compressed file" would be more informative.
Answer: done
Line 15 – Reword for clarity
Answer: done, see above.
Line 17 – The terms “focused,” “succinct,” “flexibility,” and “stereotype elements” are all terms with precise meaning that should be defined, both with respect to information theory and with respect to biological relevance.
Answer:
We added the following (lines 70-79): To describe behavior, we use the same notions as in [26]: we denote elementary movements and postures as minimal units of behavior (‘‘behavioral elements’’, for brevity), we call a ‘‘behavioral sequence’’ an arbitrary sequence of successive behavioral elements, and a ‘‘behavioral stereotype’’ – a relatively stable chain of behavioral elements. Note that the behavioral stereotype may be part of a behavioral sequence. If the variability of a stereotype is low, and it is possible with high probability to predict the appearance of each element in it, then the stereotype elements are stably related to each other. We call such a stereotype succinct, and from the ethological point of view, it may qualify as rigid and deterministic. The higher the variability of the stereotype, the less likely we are to predict transitions from one element to another. We call such a stereotype flexible.
Line 32 – What are “signal activities?” There are several other papers using the entropy to analyze many aspects of animal behavior, including
§ Stress response in tuna: https://doi.org/10.1371/journal.pone.0028241
§ Predator evasion: https://doi.org/10.1038/s41467-017-00373-2
§ Division of labor: https://doi.org/10.1007/s00040-007-0923-z
Answer:
We added the following references (lines 40-43: division of labor within ant communities (Gorelik, 2007), modifications of behavior under the stress conditions (Kadota et al., 2011), modifications of escape trajectories and predator evasion abilities in rodents (Moore et al., 2017), food-caching sequences in canine species (Gadbois et al., 2015), and so on.
Line 43 – What are “regularities?” This term should be precisely defined at its first use in the text. An example of how regularities in behavior are biologically relevant would be helpful to justify this method.
Answer: We added the following notion (lines 56-63)
By regularity we mean any characteristic of a text that makes it more predictable, such as repeating subsequences, letters or sequences that are more likely to occur when preceded or followed by certain other letters or subsequences, and so on. In general, regularities may be complex and involve arbitrary computable functions, but when applying a data compressor we restrict ourselves to those regularities it can capture. When comparing ethograms of different species as biological “texts”, the method implies the presence of regularities within behavioral patterns and finds all differences in the complexity of the ethograms. If we see such differences, then additional biological data are needed to explain them.
Line 51 – This example of genetic data is not necessary
Answer: done
Line 67 – What is meant by the phrase “reason precisely?”
Answer: removed
Line 68 – This is the first time that a “difference in complexity” is stated as the method of comparison. Please introduce and explain this earlier.
Answer:
We added: We analyze ethograms as “biological texts”, that is, as sequences of symbols from a finite alphabet [Ryabko et al., 2013]. (lines 46-47)
and: To compare the behavior of different species one needs to identify specific patterns in sequences of behavioral acts. (lines 48-49)
Line 82 – Rewording for clarity is suggested here.
Answer: removed
Line 87 – Animal kingdom doesn’t need to be capitalized
Answer: done
Lines 99-106 – These are results that should not be in the introduction
Answer: done
Lines 109-124 – The description of how each animal was captured was confusing. I would recommend stating the capture method as the species was introduced. For example, "we used 81 (41 females and 40 males) lab-reared non-pedigree adult Norway Rats (Rattus norvegicus),” etc
Answer: done
Line 122 – How are reactions “spontaneous?”
Answer: removed
Line 133 – The phrase “exemplars of” is not needed here
Answer: done
Line 137 – This definition of “stereotype” should be moved to the introduction
Answer: done
Line 140 – Change “video records” to “video recordings”
Answer: done
Line 158 – “an assessment of certain characteristics of a text” is vague. For which characteristics are you searching?
Answer: We re-wrote this
Line 163 – To which probabilistic approach are the authors referring?
Answer: removed
Line 179-184 – Move the section “So we can use the compression ratio… …is the resulting method” to the introduction
Answer: sorry, we do not agree with this.
Line 193 – “one that cannot spot fewer regularities” is difficult to parse due to double negatives. Suggest rewording for clarity.
Answer: done
Lines 198-201 – Move these numbers to the results.
Answer: done
Line 204 – This line refers to three species, whereas nine species were considered in this study.
Answer: done
Line 206 – Please provide more details regarding the “custom program.”
Answer:
We added (lines 195-197): To obtain sampled data files of equal sizes, we wrote a program which randomly chose sequences from the raw data (Figure 7, see Appendix A). We then exported them, also blank-separated, into new sampled data text files, 300 bytes in size each. There is a link to the custom program in the Appendix.
Line 223 – The manuscript would benefit from separate subsections for the results of comparison across adult rat species and the ontogenetic comparison within rats
Answer: done
Line 316 – The comparison to the grasshopper mouse should be motivated and more thoroughly described.
Answer: done
Lines 335-339 – “The proportion of… …ended in failure” should be moved to results
Answer: done
Lines 358-368 –All of the conclusion is present, nearly word for word, in the discussion. The manuscript does not benefit from this repetition.
Answer: fixed
p { text-indent: 0.49in; margin-bottom: 0.1in; direction: ltr; color: rgb(0, 0, 10); line-height: 120%; text-align: left; }p.western { font-family: "Times New Roman", serif; font-size: 16px; }p.cjk { font-size: 16px; }

Round 2
Reviewer 1 Report
p.p1 {margin: 0.0px 0.0px 5.0px 0.0px; line-height: 15.0px; font: 13.3px Arial; color: #000000; -webkit-text-stroke: #000000} p.p2 {margin: 0.0px 0.0px 5.0px 0.0px; line-height: 15.0px; font: 13.3px Arial; color: #000000; -webkit-text-stroke: #000000; min-height: 15.0px} span.s1 {font-kerning: none}I reviewed the original submission of the paper. The authors have adequately addressed the points that I raised. The paper is much improved, and the technique and findings remain interesting.
However, when conducting my original review I did not directly compare this manuscript with that of Reznikova et al., (2017). I see now that even after revisions at the request of the other reviewer of the original submission, there remains considerable overlap with this earlier paper. In particular, the section on 'hypothesis testing' and ‘Constructing sequences for hypothesis testing’ is essentially the same.
Consequently my evaluation of the ‘originality / novelty’ of this paper has dropped from ‘high’ to ‘low’. This is an important issue that I think needs addressing more directly if the paper is to be accepted.
Nevertheless, I see merit in the new application of these techniques to a larger dataset, which as the authors explain in the rebuttal includes six new species, and the newly reported observations remain interesting.
I would therefore recommend that the authors should at the very least include a very clear statement that explicitly draws attention to the fact that this section of the methods follows very closely the explanation provided in the earlier paper, and which justifies the choice to duplicate the explanation in terms of aiding the readability of the paper.
Minor issues:
line 121 - repeated sentence
line 218 - I cannot parse this sentence - please revise
line 330 - I cannot parse this sentence - please revise
Author Response
reviewed the original submission of the paper. The authors have adequately addressed the points that I raised. The paper is much improved, and the technique and findings remain interesting.
However, when conducting my original review I did not directly compare this manuscript with that of Reznikova et al., (2017). I see now that even after revisions at the request of the other reviewer of the original submission, there remains considerable overlap with this earlier paper. In particular, the section on 'hypothesis testing' and ‘Constructing sequences for hypothesis testing’ is essentially the same.
Consequently my evaluation of the ‘originality / novelty’ of this paper has dropped from ‘high’ to ‘low’. This is an important issue that I think needs addressing more directly if the paper is to be accepted.
Nevertheless, I see merit in the new application of these techniques to a larger dataset, which as the authors explain in the rebuttal includes six new species, and the newly reported observations remain interesting.
I would therefore recommend that the authors should at the very least include a very clear statement that explicitly draws attention to the fact that this section of the methods follows very closely the explanation provided in the earlier paper, and which justifies the choice to duplicate the explanation in terms of aiding the readability of the paper.
Answer: we added the following fragment in the text (lines 154-155): This section of the methods follows very closely the explanations provided in the corresponding sections of [30].
Minor issues:
line 121 - repeated sentence
done
line 218 - I cannot parse this sentence - please revise
done
line 330 - I cannot parse this sentence - please revise
done
p { margin-bottom: 0.1in; direction: ltr; color: rgb(0, 0, 0); line-height: 120%; }p.western { font-family: "Liberation Serif", "Times New Roman", serif; font-size: 12pt; }p.cjk { font-family: "WenQuanYi Micro Hei"; font-size: 12pt; }p.ctl { font-family: "Lohit Devanagari", "Times New Roman"; font-size: 12pt; }
Reviewer 2 Report
The authors made substantial revisions to the manuscript in response to the previous round of comments. There are still a few issues that require clarification.
Learning: What does complexity have to do with learning? Honing a skill may reduce the number of extraneous behavioral elements associated with the skill. However, the skill itself may become more complex with time and include more behavioral elements. The authors clearly show that the number of key elements associated with hunting behaviors decreases with rat age, despite equal complexity in young and old rats. The authors state that this may be the result of optimizing prey-catching skills. Isn't this a form of learning? Perhaps the adult rats have complex behaviors because they are bored and playing with the prey, since they require few attempts to successfully subdue prey. Thus, the statements in lines 21-22 and 368-370 are not sufficiently supported by the results.
Behavioral "stereotype:" Lines 70-79 are helpful in describing the different aspects of this experiment, but would benefit from more thorough description and examples. A behavioral sequence is discussed in the methods as being separated by blanks, which is easy to follow. However, there is no description regarding how behavioral stereotypes are identified or determined. How stable must a chain of behavioral elements be to be considered a stereotype? The note is also confusing: when is a behavioral stereotype not part of a behavioral sequence? If the stereotype has stable behavioral elements, how can it have variability? Furthermore, in Table 1, should the column label "Numbers of stereotypes in raw data text files" be instead called the number of blank-separated sequences? The paragraph referring to Table 1 only refers to sequences, and does not refer to stereotypes.
Why not bootstrap this sampling procedure to find every unique combination of sequence in each file? That way the resulting ratio could show a confidence interval, rather than be subject to a single randomly selected order.
The description of the significance-indicating letters in most of the figures has improved. However, this schema is still confusing in Figure 4. The confusion arises from letters indicating both comparison across species and across elements. Perhaps there is a better way to display these relationships?
Line 274: Why is this worth noting?
Author Response
Learning: What does complexity have to do with learning? Honing a skill may reduce the number of extraneous behavioral elements associated with the skill. However, the skill itself may become more complex with time and include more behavioral elements. The authors clearly show that the number of key elements associated with hunting behaviors decreases with rat age, despite equal complexity in young and old rats. The authors state that this may be the result of optimizing prey-catching skills. Isn't this a form of learning? Perhaps the adult rats have complex behaviors because they are bored and playing with the prey, since they require few attempts to successfully subdue prey. Thus, the statements in lines 21-22 and 368-370 are not sufficiently supported by the results.
Answer:
we added the following phrases:
The levels of complexities of hunting stereotypes in young and adult rats are similar, and both groups had no experience with the prey, so one can assume... (line 21)
Bites and grapples become more accurate with age in rats, even though both adult and young animals in our experiments had no prior experience with the prey. (lines 319-320)
It would seem that having more key elements and less noise ones in the stereotypes of young rats should lead to a decrease in their complexity, like it was demonstrated in “basic” ants’ stereotypes [26, 29] (lines 321-322)
The patterns of muscular activity are improved with age without any experience, and only in minor details. The levels of complexity of hunting stereotypes do not differ in young and adult rats. In sum, one can assume... (lines 374-376)
Behavioral "stereotype:" Lines 70-79 are helpful in describing the different aspects of this experiment, but would benefit from more thorough description and examples. A behavioral sequence is discussed in the methods as being separated by blanks, which is easy to follow. However, there is no description regarding how behavioral stereotypes are identified or determined. How stable must a chain of behavioral elements be to be considered a stereotype? The note is also confusing: when is a behavioral stereotype not part of a behavioral sequence? If the stereotype has stable behavioral elements, how can it have variability?
Answer:
To meet this concern, we added the following phrases:
Note that the behavioral stereotype may be part of a behavioral sequence, and may coincide with it completely. (lines 74-75)
The stereotypes of courtship and of territorial defense can serve as good examples here, as well as hunting stereotypes of highly specialized predators [29,36]. The higher the variability of the stereotype, the less likely we are to predict transitions from one element to another. We call such a stereotype flexible, and suggest optional hunting behavior in rodents as a good model. (lines 79-82)
Furthermore, in Table 1, should the column label "Numbers of stereotypes in raw data text files" be instead called the number of blank-separated sequences? The paragraph referring to Table 1 only refers to sequences, and does not refer to stereotypes.
Answer:
we changed stereotypes to sequences in Table 1.
Why not bootstrap this sampling procedure to find every unique combination of sequence in each file? That way the resulting ratio could show a confidence interval, rather than be subject to a single randomly selected order.
Answer:
All combinations would be too many. Selecting several ones is something we could do in the future, though the statistical significance of the results is already clear as it is.
The description of the significance-indicating letters in most of the figures has improved. However, this schema is still confusing in Figure 4. The confusion arises from letters indicating both comparison across species and across elements. Perhaps there is a better way to display these relationships?
Answer:
The use of letters to indicate significant differences is a common practice in biological articles.
Line 274: Why is this worth noting?
Answer: we deleted these words
pre { direction: ltr; color: rgb(0, 0, 0); }pre.western { font-family: "Liberation Mono", "Courier New", monospace; }pre.cjk { font-family: "Nimbus Mono L", "Courier New", monospace; }pre.ctl { font-family: "Liberation Mono", "Courier New", monospace; }p { margin-bottom: 0.1in; direction: ltr; color: rgb(0, 0, 0); line-height: 120%; }p.western { font-family: "Liberation Serif", "Times New Roman", serif; font-size: 16px; }p.cjk { font-family: "WenQuanYi Micro Hei"; font-size: 16px; }p.ctl { font-family: "Lohit Devanagari", "Times New Roman"; font-size: 16px; }